# The Characterization and Amoxicillin Adsorption Activity of Mesopore CaCO$_3$ Microparticles Prepared Using Rape Flower Pollen

**Lvshan Zhou** [1,2]**, Tongjiang Peng** [1,]*****, Hongjuan Sun** [1]**, Xiaogang Guo** [3] **and Dong Fu** [2]

[1]  Key Laboratory of Ministry of Education for Solid Waste Treatment and Resource Recycle, Institute of Mineral Materials & Application, Sichuan Engineering Lab of Nonmetallic Mineral Powder Modification & High-quality Utilization, Center of Forecasting and Analysis, Southwest University of Science and Technology, Mianyang 621010, China; zhoulvshan@126.com (L.Z.); sunhongjuan@swust.edu.cn (H.S.)

[2]  School of Chemistry and Chemical Engineering, Eastern Sichuan Sub-center of National Engineering Research Center for Municipal Wastewater Treatment and Reuse, Sichuan University of Arts and Science, Dazhou 635000, China; fudongemail@163.com

[3]  College of Chemistry and Chemical Engineering, Yangtze Normal University, Chongqing 408100, China; gxg_cqu@hotmail.com

*****  Correspondence: tjpeng@swust.edu.cn; Tel.: +86-0816-2419276

**Abstract:** A precipitation reaction method was employed to prepare mesopore calcium carbonate (CaCO$_3$) using rape flower pollen as the template. CaCO$_3$ adsorbent was characterized using X-ray diffraction (XRD), scanning electronic microscopy (SEM), and Brunner−Emmet−Teller measurements (BET). The equilibrium adsorption data on amoxicillin were explained using Langmuir, Freundlich, and Temkin adsorption isotherm models. The pseudo-first order, second order, pseudo-second order, and intra-particle diffusion kinetic models were used to explore adsorption kinetics. Equilibrium adsorption of as-prepared CaCO$_3$ was better depicted using the Langmuir adsorption model with an R$^2$ of 0.9948. The separation factor ($R_L$) was found to be in the range of $0 < R_L < 1$, indicating the favorable adsorption of amoxicillin. The adsorption capacity of mesopore CaCO$_3$ reached 13.49 mg·g$^{-1}$ in 0.2 g·L$^{-1}$ amoxicillin solution. The values of adsorption thermodynamic parameters ($\Delta H^\theta$, $\Delta S^\theta$, $\Delta G^\theta$) were obtained. In addition, the adsorption process turned out to be endothermic and spontaneous for the CaCO$_3$ product at 298 K, 308 K, and 318 K.

**Keywords:** calcium carbonate; mesopore; amoxicillin; adsorption; kinetics; thermodynamics

## 1. Introduction

Calcium carbonate is an environmentally friendly and widely used material [1]. In nature, limestone, marble, and chalk are widely disseminated. Calcite, aragonite, and vaterite are three main kinds of anhydrous crystalline form [1]. Porous calcium carbonate is a modern multi-functional material and has developed quickly in last decade. An imposing specific surface area, good biocompatibility and non-toxic are the obvious features of porous calcium carbonate. Therefore, porous calcium carbonate is widely applied into all walks of life, such as drug carriers [2–4], bioceramics [5,6], biomicrocapsules [7,8], biosensors [9], bone repair [10], biomimetic mineralization [6,11], and so on. Generally, material structures and crystal forms, affected strongly by preparation and operation conditions, have rather significant effects on their application. Therefore, the fabrication of porous calcium carbonate has been a promising area of research. Up until now, the template-assisted method [12–14], mainly based on surfactants, polymers, and natural plant ingredients as a template agent, has become the mainstream method of porous calcium carbonate preparation. At the same time, coprecipitation [15], emulsion

liquid membrane [16], solvent/hydrothermal method [17], and other methods have been improved, providing more preparation references for calcium carbonate.

Antibiotics are produced in bulk and used all over the world. A considerable amount of antibiotic wastewater is drained into rivers and lakes. Amoxicillin, a kind of beta-lactam-containing antibiotic, has been widely used in both human and veterinary medicine to treat infections caused by Gram-positive or Gram-negative bacteria. Though amoxicillin exhibits a short half-life, studies found that the amoxicillin degradation products remained toxic and may become more dangerous [18]. Thus, coagulation, adsorption, biofilm, and advanced oxidation were put forward and applied in order to effectively solve the antibiotic pollution. Of all the treatment methods, the adsorption process is highly beneficial due to the advantage of not being toxic or having the ability to degrade contaminants. Nevertheless, many adsorbents are difficult to recycle and reuse. In 2014, Shinya Yamanaka [19] and Kai Yin Chong [20] reported that porous calcium carbonate was used to treat formaldehyde vapor and disodium 3,3′-[biphenyl-4,4′-diyldi(E)diazene-2,1-diyl]bis(4-aminonaphthalene-1-sulfonate) (Congo red) solution, and the adsorption capacity of porous calcium carbonate was demonstrated to be excellent. It is reported that the structure of the rape flower pollen is a regular ellipsoidal shape with uniform particle size, a transparent porous structure on the surface, and the pores have high openness [21]. However, porous calcium carbonate used to treat antibiotic wastewater, being feasible based on these achievements, has not been reported.

In this work, rape flower pollen as a template agent was employed to synthesize mesopore calcium carbonate from eggshells at a specified temperature. The structure and morphology of calcium carbonate samples were characterized using XRD, SEM, and BET. The data for synthetic calcium carbonate adsorbing amoxicillin in an aqueous solution was studied using adsorption isotherms, thermodynamics analysis, and kinetic modeling with the aim of improving the data regarding its application.

## 2. Materials and Methods

### 2.1. Materials

All reagents except amoxicillin and eggshells were analytical reagents and supplied by Chengdu Kelon Chemical Reagent Co., Ltd., Chengdu, China. Amoxicillin (minimum 99.5%) was purchased from Shanghai Aladdin Biochemical Technology Co., Ltd., Shanghai, China. Eggshells were collected from the markets of Sichuan province in China. Distilled water was used for the preparation of all test solutions.

### 2.2. Characterization

The phase of the obtained calcium carbonate was determined using powder X-ray diffraction (D8 Advance, BRUKER AXS, Karlsruhe, Germany) with a range scan of $10° < 2\theta < 90°$. The estimated standard uncertainty of the $2\theta$ measurement was $0.01°$. The morphological features were determined using SEM (MERLIN Compact, Carl Zeiss AG, Oberkochen, Germany). The specific surface area, pore size, and other parameters were obtained using BET (V-Sorb 2800TP, Gold APP Instrument Corporation, Beijing, China).

## 3. Experimental

The waste eggshell as a calcium resource was used to produce calcium carbonate microparticles using rape flower pollen as a template at room temperature. The template containing the calcium carbonate microparticles was roasted to obtain a porous adsorbent for adsorbing amoxicillin in an aqueous solution. The flow chart is given in Figure 1.

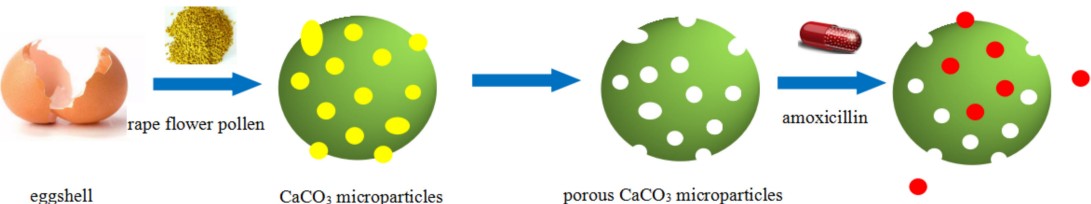

**Figure 1.** The flow chart of porous CaCO₃ preparation and adsorption.

## 3.1. Calcium Carbonate Fabrication

Mesopore calcium carbonate microparticles were prepared using eggshells. Before the experiment, a series of processes were carried on the eggshells as follows. First, the eggshells were cleaned using distilled water and the membranes were artificially removed. Then, the treated eggshell was dried in a drying oven at 105 °C for 120 min. Finally, an eggshell powder was obtained via grinding. The purchased rape flower pollen was dried at 50 °C in a drying oven for 24 h. The 0.1 g·L$^{-1}$ solution was prepared by adding 0.05 g dry rape flower pollen in 500 mL distilled water. Five grams of eggshell powder was dissolved using 75 mL 10% (volume ratio) hydrochloric acid. After filtering, 4.4 g ammonium carbonate and 2 mL 0.1g·L$^{-1}$ rape flower pollen aqueous solution were added into the filtrate to react for 45 min. At last, white precipitation was collected via filtration and dried at 105 °C for 60 min, and roasted at 500 °C for 90 min.

## 3.2. Amoxicillin Adsorption Experiment

A batch of amoxicillin adsorption experiments were carried out by adding 0.5 g prepared calcium carbonate with strong stirring for different contact times in a 200 mL amoxicillin aqueous solution under darkness for 30 min; the amoxicillin initial concentration varied from 0.01 to 0.2 g·L$^{-1}$, and the reaction temperature was 298 K, 308 K, or 318 K. The chemical oxygen demand (COD) values were determined using a potassium permanganate method for the amoxicillin solution before and after adsorption, which was filtered using a 0.45 μm filter membrane. The adsorption capacity (Γ) was calculated using the followed formula:

$$\Gamma(\mathrm{mg} \cdot \mathrm{g}^{-1}) = \frac{(c_0 - c_t) \times V}{m} \tag{1}$$

where $c_0$ and $c_t$ are the concentrations of amoxicillin at initial and time $t$ of the contact time, respectively (g·L$^{-1}$); $V$ represents the volume of amoxicillin solution (mL); and $m$ is the mass of calcium carbonate (g).

## 4. Results and Discussion

### 4.1. Characterization of Calcium Carbonate (CaCO₃) Microparticles

#### 4.1.1. Composition Analysis

The crystal structures of as-prepared calcium carbonate were studied using XRD as shown in Figure 2. As compared with the standard powder diffraction pattern, the diffraction peaks of roasted and unroasted calcium carbonate belonged to vaterite (PDF#05-0586) and (PDF#33-0268). However, there is an unknown peak (29.49°) in Figure 2b, and some calcite peaks at 38.86°, 40.76°, 51.05°, 59.92°, 62.99°, and 68.71° that are extremely weak. The existence of rape flower pollen maybe affected the crystalline conversion of calcium carbonate in some aspect. When the as-synthesized product was roasted at 500 °C for 90 min, the crystal structure changed from vaterite to calcite, which showed that vaterite is unstable at high temperatures.

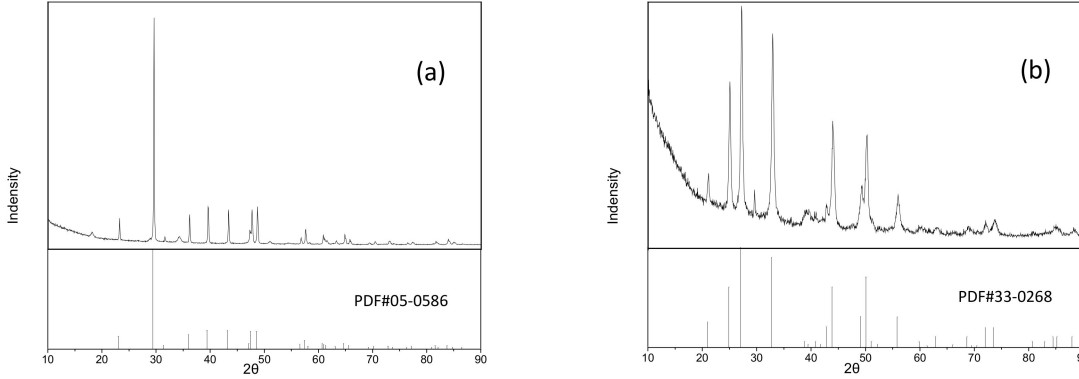

**Figure 2.** The XRD plots of calcium carbonate: (**a**) roasted calcium carbonate and (**b**) unroasted calcium carbonate.

### 4.1.2. Morphological Features

The surface morphology of products were measured using SEM images as shown in Figure 3. As seen in Figure 3a,b, products possessed a spherical morphology with diameters of 0.06–1.5 μm. After roasting, the surface of the calcium carbonate microspheres began to gelatinize and melded together. The number of internal channels became fewer, but the aperture had become larger, as shown in Figure 3c,d.

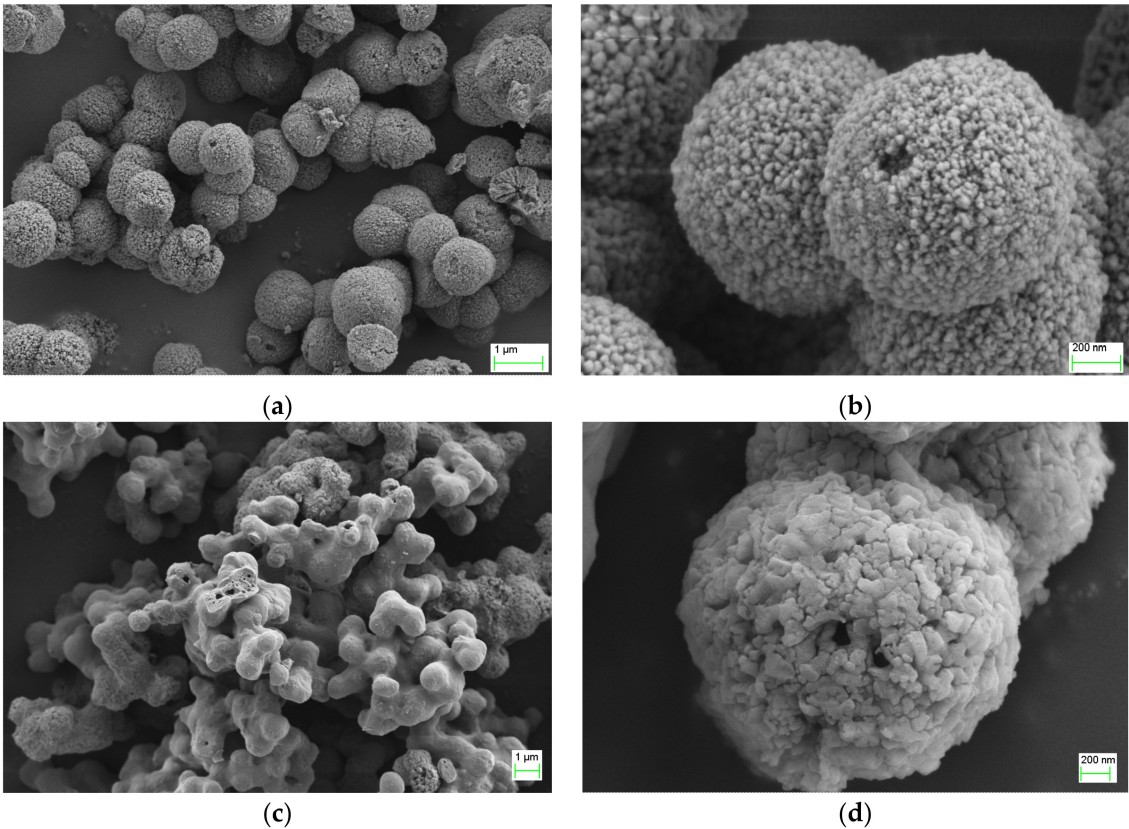

**Figure 3.** The SEM images of calcium carbonate: (**a**,**b**) unroasted calcium carbonate and (**c**,**d**) roasted calcium carbonate.

### 4.1.3. $N_2$ Adsorption/Desorption Isotherm

Pore size distribution and $N_2$ adsorption/desorption isotherms (inset) are shown in Figure 4. Based on the International Union of Pure and Applied Chemistry (IUPAC) classification, the $N_2$ adsorption/desorption isotherm of as-prepared product fits the type IV curve. The inflection point in the isotherm at low $p/p_0$ (<0.1) indicated the accomplished monolayer adsorption of $N_2$. The multilayer adsorption initiated the next stage. The adsorption hysteresis loop corresponds to the capillary condensation phenomenon of mesopores [20]. The pore size distribution showed that the diameter of the roasted product varied mainly from 3 to 15 nm (belonging to mesopores), which was consistent with the $N_2$ adsorption/desorption isotherms. However, the over 100 nm pores were rare owing to the roasting at high temperature. The BET surface area was 144.38 $m^2 \cdot g^{-1}$.

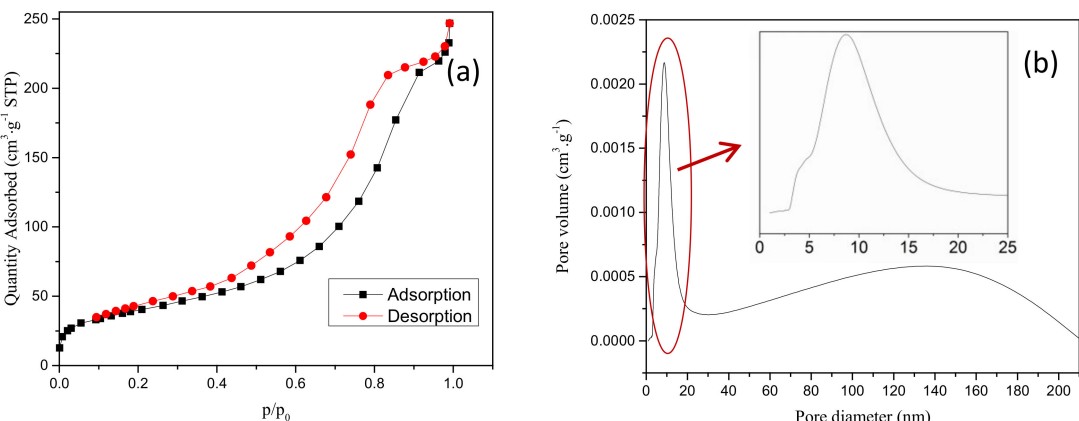

**Figure 4.** $N_2$ adsorption/desorption isotherms (**a**) and pore size distribution curves (**b**) of the roasted calcium carbonate.

### 4.2. Amoxicillin Adsorption

A series of amoxicillin standard solutions (0.01 $g \cdot L^{-1}$, 0.02 $g \cdot L^{-1}$, 0.05 $g \cdot L^{-1}$, 0.1 $g \cdot L^{-1}$, 0.15 $g \cdot L^{-1}$, 0.2 $g \cdot L^{-1}$) were prepared using an amoxicillin standard reagent. The corresponding COD was measured according to the potassium permanganate method. By plotting the COD versus $c$ values (Figure 5), the linear relationship between amoxicillin concentration and COD was y = 668.91x + 5.1268, $R^2$ = 0.9992.

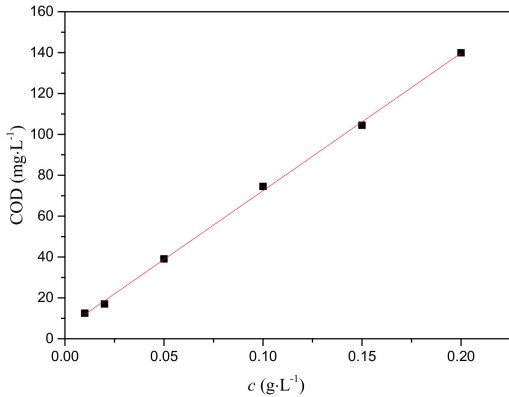

**Figure 5.** The plot of COD versus $c$ (amoxicillin concentration) for the amoxicillin standard solution.

Homemade calcium carbonate was used to deal with the amoxicillin aqueous solution. Tables 1–3 exhibit the adsorption capacity of calcium carbonate at different contact times, amoxicillin concentrations, and temperatures, respectively. As given in Table 1, the balanced adsorption amount reached 13.49 $mg \cdot g^{-1}$ after 300 min of contact time, which is higher than 333.6 $\mu g \cdot g^{-1}$ reported

in Reference [22]. Porous calcium carbonate was prepared using an microemulsion method with hexadecyl trimethyl ammonium bromide (CTAB) as a template as in Li's work [22]. In different concentration solutions, the balanced adsorption amount constantly increased with the growth of the concentration in Table 2. Meanwhile, when the increase of temperature, a similar trend appeared in a certain concentration in Table 3.

**Table 1.** The adsorption capacity of calcium carbonate at different contact times in 0.2 g·L$^{-1}$ amoxicillin solution at 298 K.

| $t$ (min) | $c_t$ (g·L$^{-1}$) | $q_t$[1] (mg·g$^{-1}$) | $t$ (min) | $c_t$ (g·L$^{-1}$) | $q_t$ (mg·g$^{-1}$) |
|---|---|---|---|---|---|
| 5 | 0.1944 | 2.23 | 10 | 0.1905 | 3.82 |
| 15 | 0.1870 | 5.20 | 20 | 0.1847 | 6.14 |
| 25 | 0.1815 | 7.42 | 30 | 0.1778 | 8.88 |
| 40 | 0.1727 | 10.92 | 50 | 0.1704 | 11.83 |
| 60 | 0.1688 | 12.49 | 90 | 0.1676 | 12.96 |
| 120 | 0.1673 | 13.07 | 150 | 0.1669 | 13.25 |
| 180 | 0.1665 | 13.39 | 240 | 0.1664 | 13.43 |
| 300 | 0.1663 | 13.49 | 600 | 0.1663 | 13.49 |

[1] The amount of adsorbate adsorbed at time $t$ of the reaction time.

**Table 2.** The adsorption capacity of calcium carbonate in different amoxicillin concentrations for 300 min at 298 K.

| $c_0$ (g·L$^{-1}$) | $c_e$ (g·L$^{-1}$) | $q_e$ (mg·g$^{-1}$) | $c_0$ (g·L$^{-1}$) | $c_e$ (g·L$^{-1}$) | $q_e$ (mg·g$^{-1}$) |
|---|---|---|---|---|---|
| 0.01 | 0.0027 | 2.91 | 0.05 | 0.0270 | 9.22 |
| 0.02 | 0.0072 | 5.11 | 0.1 | 0.0727 | 10.93 |
| 0.03 | 0.0128 | 6.89 | 0.2 | 0.1663 | 13.49 |
| 0.04 | 0.0205 | 7.80 | - | - | - |

**Table 3.** The adsorption capacity of calcium carbonate at 298 K, 308 K, or 318 K for 300 min in 0.2 g·L$^{-1}$ amoxicillin solution.

| $T$ (K) | $c_0$ (g·L$^{-1}$) | $c_e$ (g·L$^{-1}$) | $q_e$ (mg·g$^{-1}$) |
|---|---|---|---|
| 298 | 0.2001 | 0.1664 | 13.48 |
| 308 | 0.2001 | 0.1615 | 15.68 |
| 318 | 0.2001 | 0.1569 | 17.28 |

*4.3. Adsorption Isotherms*

At a specific temperature, the adsorption isotherm curve refers to the relationship between the concentration of the solute molecules in the two phases when the adsorption process reaches equilibrium. The separated substance concentration relationship in liquid and solid phase can be expressed by the adsorption equation, Langmuir adsorption isotherm (Equation (2)), Freundlich adsorption isotherm (Equation (4)), and Temkin equation (Equation (5)), which are commonly used to describe it [23,24].

$$q_e = \frac{k_L c_e}{1 + \alpha_L c_e} \text{ or } \frac{c_e}{q_e} = \frac{1}{k_L} + \frac{\alpha_L}{k_L} c_e \qquad (2)$$

where $q_e$ the amount of adsorbate adsorbed (mg·g$^{-1}$), $c_e$ is the equilibrium concentration (mg·L$^{-1}$), and $\alpha_L$ (L·mg$^{-1}$) and $k_L$ (L·g$^{-1}$) are the adsorption equilibrium constants. The theoretical monolayer saturation capacity $q_{max}$ (mg·g$^{-1}$) can be evaluated using the Langmuir equilibrium constants $\alpha_L$ (L·mg$^{-1}$) and $k_L$ (L·g$^{-1}$) using Equation (3):

$$q_{max} = \frac{k_L}{\alpha_L} \qquad (3)$$

$$q_e = k_F c_e^{\frac{1}{n}} \text{ or } \log q_e = \frac{1}{n} \log c_e + \log k_F \tag{4}$$

where $k_F$ ((mg·g$^{-1}$)·(mg·L$^{-1}$)$^{-1/n}$) and $n$ are the Freundlich constants that give the relative capacity and adsorption intensity, respectively.

$$q_e = \frac{RT}{b} \ln(A c_e) \text{ or } q_e = B \ln A + B \ln c_e \tag{5}$$

where $B = RT/b$, $B$ is the Temkin constant related to the heat of sorption (J·mol$^{-1}$), $A$ is the Temkin isotherm constant (L·g$^{-1}$), $R$ is the gas constant (8.314 J·mol$^{-1}$·K$^{-1}$), and $T$ is the temperature (K).

In this work, the adsorption capacities of the as-synthesized mesopore calcium carbonate were studied using Langmuir, Freundlich, and Temkin isotherm models for the adsorption of amoxicillin. The result from plots (Figure 6) indicate R$^2$ = 0.9948, 0.9937, and 0.9399 for Langmuir (Figure 6a), Freundlich (Figure 6b), and Temkin (Figure 6c), respectively. In terms of R$^2$, Langmuir was more favorable than Temkin and Freundlich. To determine whether the adsorption of amoxicillin on calcium carbonate was favorable, the values of $R_L$ ($R_L = 1/(1 + \alpha_L \cdot c_0)$) and $n$ for Langmuir and Freundlich were evaluated. The results indicate that $R_L = 6.94 \times 10^{-4}$ and $n = 2.8$. The work of N.K. Goel et al. [24] reported that if $R_L$ is less than unity, then adsorption is favorable. Hence, the as-synthesized mesopore calcium carbonate is a good adsorbent for amoxicillin.

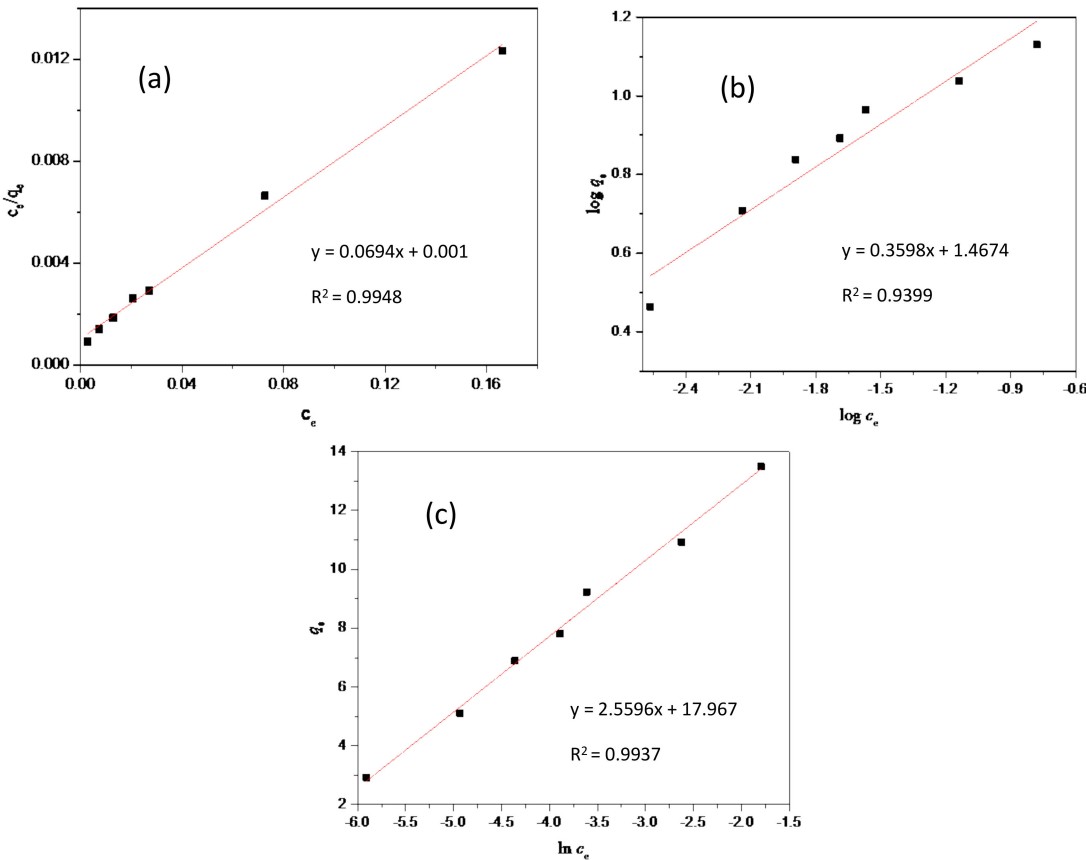

**Figure 6.** Adsorption isotherms of amoxicillin using calcium carbonate: (**a**) Langmuir isotherm, (**b**) Freundlich isotherm, and (**c**) Temkin isotherm.

## 4.4. Thermodynamic Parameters Analysis

A series of experiments on temperature dependence of as-prepared calcium carbonate were carried out to obtain thermodynamic adsorption parameters, such as standard Gibbs free energy ($\Delta G^\theta$,

J·mol$^{-1}$), enthalpy ($\Delta H^{\theta}$, J·mol$^{-1}$), and entropy ($\Delta S^{\theta}$, J·mol$^{-1}$·K$^{-1}$) [24], which can be determined using Equations (6) and (7):

$$\ln k_D = \frac{\Delta S^{\theta}}{R} - \frac{\Delta H^{\theta}}{RT} \tag{6}$$

$$\Delta G^{\theta} = \Delta H^{\theta} - T\Delta S^{\theta} \tag{7}$$

where $k_D$ is the distribution coefficient of the adsorbent and is equal to $q_e/c_e$, $T$ is the absolute temperature, and $R$ is the gas constant. The plot of ln$k_D$ versus $1/T$ (figure is not shown) is linear with the slope and the intercept giving values of $\Delta H^{\theta}$ and $\Delta S^{\theta}$.

The values of adsorption thermodynamic parameters are listed in Table 4. The enthalpy change is positive implying that the adsorption process is endothermic. The negative value of $\Delta G^{\theta}$ confirms the feasibility of the adsorption process and also indicates spontaneous adsorption of amoxicillin on the product.

**Table 4.** Thermodynamic parameters for the adsorption of amoxicillin on as-prepared calcium carbonate.

| $T$ (K) | $k_D$ | $\Delta H^{\theta}$ (J·mol$^{-1}$) | $\Delta S^{\theta}$ (J·mol$^{-1}$·K$^{-1}$) | $\Delta G^{\theta}$ (J·mol$^{-1}$) |
|---|---|---|---|---|
| 298 | 81.01 | - | - | −10,897.10 |
| 308 | 96.02 | 12,108.51 | 77.20 | −34,674.69 |
| 318 | 110.13 | - | - | −59,224.29 |

## 4.5. Adsorption Kinetics

The kinetics of adsorption was investigated using the pseudo-first order, second order, pseudo-second order, and intra-particle diffusion model, as represented in Equations (8)–(11):

$$\text{Pseudo-first-order model } \ln(q_e - q_t) = -kt + \ln q_e \tag{8}$$

$$\text{Second-order model } \frac{1}{q_e - q_t} = kt + \frac{1}{q_e} \tag{9}$$

$$\text{Pseudo-second-order model } \frac{t}{q_t} = \frac{t}{q_e} + \frac{1}{kq_e^2} \tag{10}$$

$$\text{Intra-particle diffusion model } q_t = kt^{0.5} + C \tag{11}$$

According to the data in Table 1, linear plots were drawn based on the kinetics models mentioned above, as shown in Figure 7. It is apparent that the pseudo-second-order model was more favorable in terms of R$^2$, showing that the adsorption phenomenon followed the second order. The intra-particle diffusion plot (Figure 7d) indicates multi-linearity represented by two different stages, suggesting involvement of two different adsorption processes, which is similar to Reference [24]. In the first and fast-changing stage, the mass transfer through the boundary layers of liquid and adsorption on the available occurred on the external surface of the adsorbent of amoxicillin. After this stage, the amoxicillin molecules entered the interior of the adsorbent through the porous structures, and eventually were adsorbed on the internal active sites. Owing to increasing diffusion resistance of the amoxicillin molecule transportation from the external surface to the bulk of the adsorbent, the adsorption rate slowed down, which is demonstrated in the second stage. However, the fitted straight line that does not pass through the origin displays that intra-particle diffusion was not the only step controlling the adsorption process [23,24].

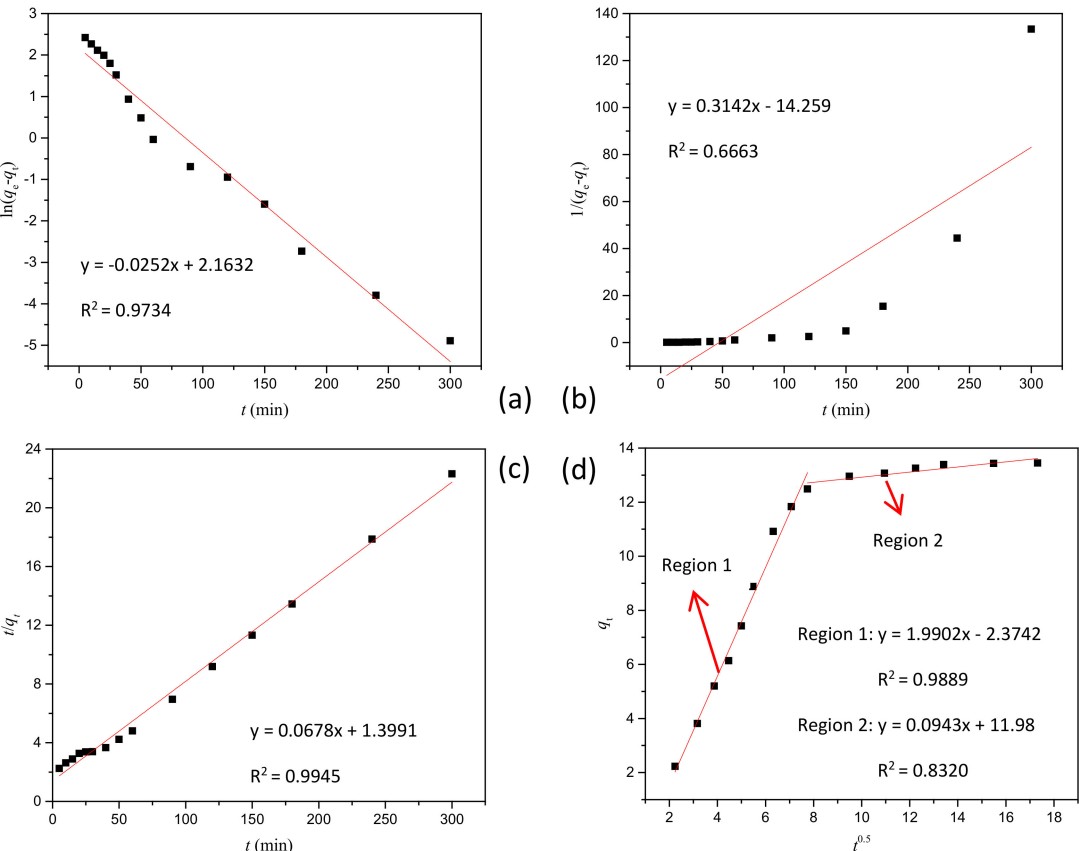

**Figure 7.** The plot of different kinetics models: (**a**) pseudo-first-order model, (**b**) second-order model, (**c**) pseudo-second-order model, and (**d**) intra-particle diffusion model.

## 5. Conclusions

Mesopore calcium carbonate with a BET surface area of 144.38 $m^2 \cdot g^{-1}$ was produced successfully using rape flower pollen as a template for removing amoxicillin in aqueous solution. The results indicated that it is a favorable adsorbent, and its adsorption capacity reached 13.49 $mg \cdot g^{-1}$. Langmuir, Freundlich, and Temkin isotherm models were used to interpret the adsorption phenomenon of the adsorbent. The Langmuir isotherm was the best fitting model for the equilibrium adsorption data gathered from the product. For low pressure ($p/p_0 < 0.1$), the monolayer adsorption was dominant, and in the following stage, multilayer adsorption dominated instead. Adsorption kinetics was demonstrated to follow a pseudo-second-order model. The negative value of $\Delta G^\theta$ signified that the adsorption reaction was a spontaneous adsorption, and the positive value of $\Delta H^\theta$ indicated the adsorption process was endothermic.

**Author Contributions:** L.Z. contributed to the literature search, study design, experimental study, data analysis, and writing—review and editing; T.P. contributed to the design of the work and data analysis; H.S., X.G., and D.F. contributed to the work of writing—review and editing.

**Funding:** This research was funded by the Opening Project of Key Laboratory of Green Catalysis of Sichuan Institutes of Higher Education (Grant No. LYJ1605); the program projects funded of Sichuan University of Arts and Science (Grant No.2018SCL001Y) and the Scientific and Technological Innovation Team Foundation of Southwest University of Science and Technology, China (Grant No.17LZXT11).

**Conflicts of Interest:** The authors declare no conflict of interest.

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
