# Peer review of "The Characterization and Amoxicillin Adsorption Activity of Mesopore CaCO3 Microparticles Prepared Using Rape Flower Pollen"

_minerals, doi:10.3390/min9040254_

Round 1
Reviewer 1 Report
Following an overall inquiry into the reviewed article, I consider it to be a very interesting investigation of the fabrication, characterization and adsorption activity of mesopore CaCO3 microparticles.
The manuscript contains interesting and valuable data, which have been mostly correctly evaluated and interpreted. Organization and clarity of the manuscript is also generally good.
Overall Recommendation: Accept in present form
Author Response
Dear reviewer,
We are writing to express our sincere gratitude for your comments in our manuscript entitled as “The characterization and amoxicillin adsorption activity of mesopore CaCO3 microparticles prepared using rape flower pollen (minerals-484327)” .
Firstly, please accept our humblest apologies to you and reviewers for our carelessness and some low-level mistakes. Abundant thanks to you for your patience, expertise and genuine concerns.
Secondly, according to your requirements, we rewrite the manuscript thoroughly and have resubmitted it as a new submission after careful modification and errors checking corresponding to comments proposed by you and reviewers. It's really extremely thankful for you to pay attention to our paper in your busy schedule. And please let us know whether you think the manuscript needs to be improved at your convenience.
We sincerely hope you can give me a chance of my paper published in minerals. We wish you success in work and everything goes well in the coming days.

Reviewer 2 Report
In the manuscript submitted by Lvshan Zhou and co-authors, the authors prepared mesopore CaCO3 microparticles using rape flower pollen as the template. Then, the authors investigated the adsorption characteristics of amoxicillin on the material. The content of this paper is within the scope of minerals. Comments are listed below.
1. Add the terms “rape flower pollen” and “amoxicillin” in title.
2. Add the term “amoxicillin” in line 15-18 in abstract.
3. Why did the authors use rape flower pollen as the template? And why did the authors choose amoxicillin among antibiotics? Describe the motivations in introduction section.
4. How did the authors obtain rape flower pollen aqueous solution? Add more detail in the section 3.1.
5. In the section 3.2, did the authors conduct filtration or centrifugation (i.e., solid-liquid separation) before COD measurements?
6. Compare the adsorption performance of the developed material with previously reported adsorbents for amoxicillin.
Author Response

(The authors gave the same response as above.)

Reviewer 3 Report
I find it hard to assess the signifcance of the adsorption data for the porous calcium carbonate without refercne data for non porous samples. The authors should carry out this control experiment or provide a refence to the work of others in which it has been carried out (perhaps reference 18 or 19?) and give the values.
The English requires significant editing and I would like to see the plots of C versus COD referred to in lines 139-141.
Author Response

(The authors gave the same response as above.)

Round 2
Reviewer 3 Report
This paper describes the adsorption properties towards amoxycillin of CaCO3 samples prepared using rape flower pollen to produce porosity. The work seems sound but the authiors should provide more detail on the results from the masters thesis in reference 23 as supporting information. English language editing is still required.
Author Response
Dear editor and reviewers,
Thanks for your letter and the reviewers’ comments concerning our manuscript entitled “The characterization and amoxicillin adsorption activity of mesopore CaCO3 microparticles prepared using rape flower pollen (minerals-484327)” . We have made carefully again modification and errors checking, including the structure of the paper, expressions, grammar etc. In new manuscript, we have added two authors- Xianggang Guo and Dong Fu, because they have sufficiently contributed to the work of writing-review and editing. This decision was approved by all the authors of this article.
The main corrections and the responds to reviewers comments are as following:
(1) Extensive editing of English language and style required .
We have made carefully again modification and errors checking, including the structure of the paper, expressions, grammar etc.
(2) This paper describes the adsorption properties towards amoxycillin of CaCO3 samples prepared using rape flower pollen to produce porosity. The work seems sound but the authiors should provide more detail on the results from the masters thesis in reference 23 as supporting information. English language editing is still required.
Thanks for your professional advice. We have adeed the detail on the results from the masters thesis in reference 23.
which is higher than 333.6 μg·g-1 reported in the reference[23]. Porous calcium carbonate was prepared by microemulsion method using hexadecyl trimethyl ammonium bromide (CTAB) as a template in Li’s work[23].